# Design and Evaluation of a Sticky Attractant Trap for Intra-Domiciliary Surveillance of *Aedes aegypti* Populations in Mexico

**DOI:** 10.3390/insects14120940

**Published:** 2023-12-11

**Authors:** Keila Elizabeth Paiz-Moscoso, Luis Alberto Cisneros-Vázquez, Rogelio Danís-Lozano, Jorge J. Rodríguez-Rojas, Eduardo A. Rebollar-Téllez, Rosa María Sánchez-Casas, Ildefonso Fernández-Salas

**Affiliations:** 1Laboratorio de Entomología Médica, Facultad de Ciencias Biológicas, Universidad Autónoma de Nuevo León, Monterrey 30700, Mexico; keila.paizm@uanl.edu.mx (K.E.P.-M.); eduardo.rebollartl@uanl.edu.mx (E.A.R.-T.); 2Centro Regional de Investigación en Salud Pública, Instituto Nacional de Salud Pública, Tapachula Chiapas 62100, Mexico; luis.cisneros@insp.mx (L.A.C.-V.); rdanis@insp.mx (R.D.-L.); 3Unidad de Patógenos y Vectores, Centro de Investigación y Desarrollo en Ciencias de la Salud, Universidad Autónoma de Nuevo León, Monterrey 30700, Mexico; jorge.rodriguezr@uanl.mx; 4Facultad de Medicina, Veterinaria y Zootecnia, Universidad Autónoma de Nuevo León, Monterrey 30700, Mexico

**Keywords:** *Aedes aegypti*, sticky trap, lure, entomological surveillance

## Abstract

**Simple Summary:**

Globally, current efforts to contain the transmission and spread of dengue, chikungunya fever, and Zika have not proven to be effective. *Aedes aegypti* is a mosquito that remains inside houses throughout the day, limiting some strategies such as outdoor spatial spraying and using nighttime nets treated or not with insecticides. It is also important to mention that vector control is completely reactive, as it is only used to respond to confirmed clinical cases that are susceptible to delays in reporting, and even worse, this activity is carried out with limited human, material, and financial resources. Therefore, there is an urgent need to reformulate current control strategies and enhance the household surveillance of adult populations to have a significant and measurable impact on infective mosquito populations and the transmission of these diseases. This study aims to design and evaluate a low-cost attractant sticky trap that provides significant surveillance results for monitoring the presence of indoor *Ae. aegypti* and relative density populations. Decision-making authorities may have access to field-realistic vectors and data from cases in time and space to establish a risk outbreak threshold and start implementing control measurements.

**Abstract:**

Surveillance consists of systematic data collection, analysis, and interpretation and is essential for planning and implementing control activities. The lack of success in the control and surveillance of the *Ae. aegypti* mosquito elsewhere demands the development of new accessible and effective strategies. This work aimed to develop and evaluate an adhesive lure trap for household indoor surveillance of *Ae*. *aegypti*. Based on a bibliographic review, four compounds that have significant attraction percentages for *Ae*. *aegypti* were considered. Our more effective blend was determined through preliminary bioassays using the high-throughput screening system (HITSS) and 90 × 90 cm mosquito cages. We designed a low-cost, pyramid-shaped, sticky cardboard trap to incorporate the selected blend. Semi-field 2 × 2 m cages and field tests were utilized to evaluate its effectiveness through mosquito capture percentages. In laboratory tests, blend number 2 presented an attraction percentage of 47.5 ± 4.8%; meanwhile, in semi-field cages, a 4-inch, 110 v powered fan was used to disperse the attractants, and then a similar capture percentage of 43.2 ± 4.0% was recorded. Results were recorded during the field evaluation of the at-house indoor environment and were compared with those recorded with the golden-standard BG-Sentinel trap, i.e., our prototype trapped an average of 6.0 ± 1.5 mosquitoes versus 10.0 ± 2.6. In most Latin American countries, there is a lack of formal and accessible strategies for monitoring adult populations of *Ae*. *Aegypti*; therefore, we must develop tools that reinforce entomological surveillance methods.

## 1. Introduction

Since the beginning of 2023, dengue outbreaks of significant magnitude have been recorded in the WHO Region of the Americas, with close to 3 million suspected and confirmed cases of dengue reported so far this year, surpassing the 2.8 million cases of dengue registered for the entire year of 2022. Added to this, all four dengue virus serotypes (DENV1, DENV2, DENV3, and DENV4) are present in the Region of the Americas [1].

Chikungunya was first detected in the region in 2013 on the island of Saint Martin, and a year later, it had spread to most countries in the region. More than one million cases were reported in the first year after its introduction to the continent. In the first four months of 2023, an increase in the circulation of chikungunya was detected in the region, with more than 214,000 cases reported. [2]. Despite the global reduction in the disease caused by a decrease in Zika virus cases since 2017, the circulation of this mosquito-transmitted virus has been confirmed in 89 countries worldwide. Although incidence levels remain low, sporadic increases have been observed in recent years [3]. Evidence supports *Aedes aegypti* as the primary vector in the Americas, and the link between the Zika virus, microcephaly, and other neurological disorders, such as Guillain–Barre syndrome, led the WHO to declare this disease a health emergency of international concern [4,5].

Dengue infection is a recurrent cause of hospitalizations, especially in children [6,7]. In February 2020, the Pan American Health Organization (PAHO) released an epidemiological alert for dengue in the Americas due to an increase in the incidence rate (81.51 cases per a population of 100,000), which was seven times more than that reported during the same period in 2019 (11.09 cases per a population of 100,000), and mortality related to DHF has increased annually, generating a significant economic burden for the health systems of Latin American countries [8,9]. 

In Mexico, as in most countries that belong to the subtropical area of the American continent, *Ae*. *aegypti* is the primary vector of dengue, chikungunya fever, and Zika viruses. Up to epidemiological week number 31 of 2023, in the country, there have been 4554 confirmed cases of dengue without alarm data and 3468 cases with alarm data, 249 cases of severe dengue, 13 deaths due to dengue infection, and 3 cases of infection by Zika virus [10]. In Mexico, the term dengue cases with alarm data refers to the fact that the patient may present with intense and continuous abdominal pain, persistent vomiting, fluid accumulation, mucosal bleeding, an altered state of consciousness, hepatomegaly, and a progressive increase in hematocrit. Likewise, the General Directorate of Epidemiology determined the age groups most affected in 2023 by dengue infections. The group of 10–14-year-olds was the most affected by nonsevere dengue, and the groups of 10–14 and 15–19-year-olds were the most affected by severe dengue [11]. 

Globally, current efforts to contain the transmission and global spread of dengue, chikungunya fever, and Zika have not proven effective. In Mexico, the Official Standard NOM-032-SSA2-2014 mentions the sustained management of vector control through chemical, biological, and physical control to avoid, to the greatest extent possible, the risk of transmission of one or more vector-borne diseases [12]. The lack of success of control strategies is likely related to the excessive and widespread use of insecticides that has caused the development of mosquitoes resistant to these products throughout the world [13] and the tendency to substitute a particular pesticide, for which resistance has been detected, with a new one (susceptible mosquito). However, evidence has been presented of the harmful effects derived from pesticide use [14]. In addition to the aforementioned issues, *Ae*. *aegypti* is a vector that remains inside houses throughout the day, limiting some of these strategies, such as spatial spraying and using bed nets treated or not with insecticides. However, it is very important to mention that vector control in Mexico is entirely reactive to delayed reports of clinical cases, which are inherent in the passive surveillance system. Additionally, this activity is carried out with limited human, material, and financial resources [15,16]. 

Therefore, it is necessary to reformulate the current *Aedes* arboviral disease control and prevention strategies. We have learned about the serious impact on disease incidence when primary health care fails to identify suspected cases, affecting confirmatory medical procedures such as laboratory diagnosis and case confirmation and causing a lack of timely deployment of vector control measures. Surveillance consists of the systematic collection, analysis, and interpretation of data, and it is essential for planning and implementing control activities on adult mosquito populations before and after outbreaks. It is also the first indicator used to evaluate the significant and measurable impact on vector populations and the transmission of these diseases [17]. This is why this work proposes the design and evaluation of a low-cost attractant sticky trap that allows for the monitoring of adult populations of *Ae*. *aegypti* in indoor house environments. The future use of field assessment data can support establishing the precise moment and the appropriate strategy to maintain an immediate and sustained reduction in vector control and case incidence. 

## 2. Materials and Methods

### 2.1. Development of the UANL Attractant Sticky Trap (UANL Aedes Trap^®^) 

Based on a literature systematic review [18], we selected individual compounds (lactic acid, hexanoic acid, ammonium chloride, and linalool) and concentrations showing significant percentages of attraction in *Ae*. *aegypti* females. Additionally, based on our experience with linalool, an aromatic oil produced by plants with repellency and attractancy properties at low concentrations [19], we prepared 12 different blends to run attraction assays using the HITSS, as explained in Section 2.2. Experimental data on the higher attraction percentages after this screening led to the selection of two blends with chosen compounds and concentrations. Blend 1: linalool, 10% (Sigma Aldrich 97%; Merck, St. Louis, MI, USA); lactic acid, 1% (CTR 88.50%; CTR Scientific, Monterrey, NL, Mexico); hexanoic acid, 0.1% (Sigma Aldrich 97%, St. Louis, MI, USA); and ammonium chloride, 0.1% (CTR 99%). Blend 2: linalool, 15%; lactic acid, 10%; hexanoic acid, 1%; and ammonium chloride, 1%. We named our prototype UANL *Aedes* Trap^®^ to protect the university’s and scientists’ property rights.

### 2.2. Attraction Bioassay Screening in the Laboratory

*Aedes Aegypti* eggs were obtained from the Insectary of the Regional Center for Research in Public Health (CRISP). For the bioassays, females (4 to 7 days old), maintained at 27 ± 2 °C and 80 ± 10% RH under a photoperiod of 12:12 light/dark, were used. Mosquitoes were maintained only with a 10% sugar solution soaked in cotton swabs. For the attraction tests of candidate compounds, the high-throughput research processing system (HITSS) was used for the initial screening. The system has a modular design that allows for the evaluation of contact irritation, spatial repellency, and toxicity of the products [19,20]. Whatman No. 2 filter papers (4 × 2 cm) impregnated with 50 µL of the two testing blends were individually placed in the treatment cylinder. Once the treatment cylinder was prepared, it was attached to the plexiglass cylinder at one end of the HITSS, and the negative control cylinder (Whatman No. 2 (4 × 2 cm)) impregnated with 50 µL of acetone (Binden 97%) was attached to the other end. The butterfly valves were kept closed at the junction with both cylinders. Fifteen mosquitoes were placed in the central compartment, and the entire system was completely covered with a dark cloth to prevent the passage of light. The mosquitoes remained in the plexiglass chamber for acclimatization for 30 s, and then the valves were opened simultaneously to expose the mosquitoes to the treatments for 20 min. After 20 min, the mosquitoes were counted in the treatment chamber (attraction), control chamber (repellent), and central chamber (no response).

Four repetitions were carried out for each blend, which included the BG trap bait (lure), considered the golden standard, as the positive control (0.50 g of the BG-Sentinel bait), and acetone as the negative control. The system was cleaned with acetone and aerated for 30 min between bioassays with different products or concentrations. The blend yielding higher attractiveness results was selected for an additional larger-flying-volume evaluation under insectary conditions.

#### Evaluation of Selected Blend 2′s Consistency in Larger Entomological Cages 

Two experiment series were carried out to validate the attraction consistency of Blend 2 in larger flying spaces: In the first, individual strips of Whatman No. 2 filter paper (4 × 2 cm) were impregnated with 100 µL of the treatments (Blend 1, Blend 2, negative control of acetone, or positive control of 5 g BG bait placed in Petri dish lids), and 20 mosquitoes were introduced into a metal cage (90 × 90 × 90 cm), where they were allowed to acclimatize for 5 min. After 30 min, the mosquitoes approaching or landing on the impregnated strips were counted and removed. In the second type of experiment, the impregnated strips were introduced alongside the use of an electric 110 V fan (speed of 1.0 m/s, 8″ diameter; Steren S.A. de C.V., Mexico City, Mexico) to favor the dispersion of the blend; similarly, 20 mosquitoes were introduced and allowed to acclimatize for 5 min. After 30 min, the mosquitoes approaching or landing on the strips were counted and removed. Each experiment lasted three hours, with observations and the introduction of new mosquitoes every 30 min. Each treatment with and without a fan was repeated for 5 days in the insectary. The laboratory was ventilated for 24 h between bioassays with different treatments.

### 2.3. Prototype Design of UANL Aedes Trap for Assays under Semi-Field and Field Conditions 

Based on the genetic traits of *Ae. aegypti* related to flying and host-seeking behavior [21], three elements were incorporated into our UANL *Aedes* trap: Firstly, we used a white-colored cardboard pyramid mold with entrance windows or “holes” near the floor edge (13 × 24 cm, with four 5 × 3 cm side windows, Mod 42 × 24, white; Multi Empaque Monterrey S.A de C.V, Monterrey, NL, Mexico). Secondly, the interior walls of the trap were lined with odorless adhesive paper (Sku Travo l0001; Tetengo S. de R.L de C.V., Santa Catarina, NL, Mexico), and the floor of the trap was lined with adhesive paper (type cat paper; Eco Company S.A., Cartago, Costa Rica). Thirdly, a second variant was integrated with a small fan (4”, 12 V, 1.44 W, current consumption: 150 mAm, version 1.3, silent function: 29.8 dBA; Electronica Steren SA. De C.V, Mexico City, Mexico) which was placed on the upper part of the trap to facilitate the suction of the mosquitoes and the dispersal of the attractant blend (Figure 1). 

#### 2.3.1. Attractant Blend 2 Cartridge

Using an Eppendorf tube (1.5 mL), 0.5 g of polyacrylate hydrogel (Green Forest Mexico©, Puebla, Mexico) was activated with water, and 1 mL of our attractant lure (Blend 2) was added. The blend was prepared 24 h before the bioassays and placed inside the trap 5 min before starting each test. Regarding mosquitoes for the assays, we used field-collected larvae of *Ae*. *aegypti* that were reared to adult stages without blood feeding in the CRISP/INSP insectary (F1-F5) until they were between 4 and 7 days old. 

#### 2.3.2. Semi-Field Assays of UANL Aedes Trap at Greenhouse Scale 

The bioassays were carried out in a large 10 × 50 × 5 m long greenhouse of the experimental field station of the Regional Public Health Research Center (CRISP-INSP) located in the Río Florido Ejido, Tapachula, Chiapas, from March to May 2023. The average ambient temperature during the bioassays was 35.6 °C, and the relative humidity was 62%. 

In order to obtain the actual capture percentage offered by the UANL *Aedes* trap alone and see if there was an improvement in the capture percentages when adding each of the factors that were included in the design (bait and fan), four types of bioassays were carried out in 2 × 2 × 2 m cages. First, the attractant sticky trap alone (AST) was placed in the cage with a plant (*Mentha spicata*) as a resting refuge and a container with 10% glucose as food for the mosquitoes. Second, the attractant sticky trap plus our selected attractant bait (Blend 2) (ASTB), a plant (*Mentha spicata*), and a container with 10% glucose were placed in the cage. Third, the attractive sticky trap with the attractant bait (Blend 2) was used alongside a fan (ASTBF), a plant (*Mentha spicata*), and a container with 10% glucose. In the fourth bioassay, the golden standard or positive control (the BG-Sentinel trap with its integrated BG bait (BG-SB)), a plant (*Mentha spicata*), and a container with 10% glucose were used (Figure 1).

Once our five-in-a-row screened cages were set, 100 females of *Ae*. *aegypti* were released, and observations were made for 10 min to ensure the acclimatization of the mosquitoes and their flight ability. The first and second readings of trapped mosquitoes were taken 24 h and 7 days after the first release, respectively. Once the 7-day reading was completed, a second release was conducted with 100 fresh female mosquitoes without changing the traps. As a last step, the last reading was taken after 24 h or day 8. Five daily repetitions were carried out for each treatment, AST, ASTB, and ASTBF, while heavy rainfall allowed us to only carry out 3 repetitions for the BG-SF. The screened cages were ventilated 24 h a day between bioassays.

#### 2.3.3. Field Evaluation of UANL Aedes Trap in Indoor Household Environments

Based on higher semi-field and field results, we evaluated and compared our UANL *Aedes* trap (ASTBF) with the standard BG-Sentinel. The bioassays were carried out in June 2023. The neighborhoods used were preselected because, according to data from the Secretary of Health of the State of Chiapas, they are within the areas that report the highest number of confirmed dengue cases each year. Before placing the traps, an informative talk about the design and use of the traps was given to the head of the family or housewife (family members who were willing to attend could participate in the talk); the next step was to request the consent of the head of the family for his house and family to participate in the field evaluation. 

Five houses were selected in different neighborhoods of the city of Tapachula, Chiapas (in the La Primavera, 5 de Febrero, Barrio Nuevo, and Libertad subdivisions). In each of them, with prior authorization and training of the head of the family, one 110 V powered UANL *Aedes* trap (ASTBF) was placed resting over the living room or kitchen chairs and similar resting surfaces at a height range of 60–80 cm. The traps were checked daily for maintenance without removing the trapped mosquitoes; after 7 days, the mosquitoes were counted and identified. Likewise, in three of the selected house groups (in Barrio Nuevo and 5 de Febrero), the powered BG-Sentinel traps (BG-SB) were placed on the floor, and they were similarly removed 7 days later for counting and identifying the trapped mosquitoes.

### 2.4. Statistical Analysis

All data generated were checked for normality assumptions and homogeneity of variance through the Shapiro–Wilks and Levine tests, respectively. The data obtained from the HITSS and from the 90 × 90 × 90 cm entomological cages, as well as the capture percentages obtained from the semi-field assays 24 h, 7 days after the first release, and 24 h after the second release, followed a normal distribution. All data were analyzed using a one-way ANOVA and a Tukey test to determine the difference between treatments with AMOVI Statistical Packages 3.2.21 (The Jamovi Project (2023) Sydney, Australia, and Bernhard Kllngenberg (https://artofstat.com/web-apps 2022)) (accessed on 16 August 2023). For the data obtained in the field, Welch’s *t*-test was calculated to compare the proportions of mosquitoes captured between the two types of traps and the sex ratios. Likewise, for the data obtained in the HITSS, the spatial activity index (SAI = ((Nc-Nt)/(Nc + Nt)) × (Nm/N)) was calculated, where Nc = the number of mosquitoes in the control cylinder, Nt = the number of mosquitoes in the cylinder treatment, and Nm = the total number of mosquitoes. Results closer to −1 indicate a greater spatial attraction to the treatment [19]. 

## 3. Results

### 3.1. Tests Using the HITSS

As seen in Figure 2, there were significant differences in the attraction percentages obtained with the HITSS. The mean attraction percentage for the negative control was 10.0 ± 4.1%, while it was 27.5 ± 2.5% for the BG-Sentinel bait positive control, 42.5 ± 4.8% for Blend 1, and the highest percentage of 47.5 ± 4.8% for Blend 2 (*F* = 16.6, *df* = 3, *p* = < 0.001). According to the SAI calculations, the negative control obtained a spatial activity index of −0.03 ± 0.00, while the positive control was −0.18 ± 0.05, Blend 1 was −0.20 ± 0.00, and Blend 2 was −0.45 ± 0.05. With the data obtained from the HITSS, Blend 2 showed the highest percentage of attraction and the highest SAI.

### 3.2. Consistency of Blend 2 Attraction in Larger Entomological Cages

Figure 3 shows that in the tests without a fan, the attraction mean was reduced, showing the lowest result of 0.83 ± 0.8% for the negative control made of acetone strips. Similarly, while the positive control BG-Sentinel bait strip’s attraction mean was 8.3 ± 3.1%, Blend 1 obtained a mean of 9.2 ± 3.5%, and Blend 2 showed the highest performance at 15.8 ± 4.4% without fan air dispersion. Conversely, in the tests where a fan was used, the negative control yielded a 0.0 ± 0.0% attraction mean, the positive control yielded a 34.8 ± 5.6% mean, Blend 1 produced a 28.3 ± 4.8% mean, and Blend 2 yielded the highest mean of 40.8 ± 4.6% (*F* = 34.9, *df* = 5, *p* = < 0.001). With the data obtained through the open monitoring system provided by the 90 × 90 cm × cages, it was concluded that Blend 2, helped by the fan, was the mix that presented the best percentage of attraction, which was followed by the gold-standard BG-Sentinel bait with an integrated fan.

### 3.3. Semi-Field Evaluation of UANL Aedes Trap at Greenhouse Scale

The results from the larger 2 × 2 × 2 m screened cages changed on the side of the BG-SB golden-standard trap. The mean capture percentages from the semi-field tests 24 h after the first release of 100 female mosquitoes were 9.6 ± 2.7% for the AST, 21.6 ± 3.5% for Blend 2 or the ASTB, and 43.2 ± 4.0% for the ASTBF, and the highest result was 63.7 ± 11.1% for the BG-SB with the fan system (*F* = 17.2, *df* = 3, *p* = 0.002) (Table 1). The capture means recorded 7 days after the first release were 12.1 ± 3.3% for the AST, 27.0 ± 5.8% for the ASTB, and 64.3 ± 11.6% for the BG-SB (*F* = 13.7, *df* = 2, *p* = 0.001) (Table 1). Due to adverse weather conditions causing potential damage to the traps, the capture percentages were not recorded for 7 days after the first release for the ASTBF trap. Finally, the capture percentages in the semi-field tests 24 h after the second release were 2.4 ± 0.8% for the AST, 10.0 ± 1.8% for the ASTB, 40.8 ± 2.9% for the ASTBF, and 65.3 ± 7.5% for the BG-SB (*F* = 13.77, *df* = 3, *p* = 0.0028) (Table 1). With the data obtained from the semi-field tests, it can be concluded that the BG-Sentinel trap with the bait and integrated fan was the one that presented the highest attraction percentages 24 h and 7 days after the first release and 24 h after the second release.

### 3.4. Field Evaluation of UANL Aedes Trap 

Only our ASTBF trap was evaluated in the field and placed in the living room or bedroom of the selected houses during a 7-day sampling period. Dwellers complained about the BG bait odor, so we decided to relocate these traps outside at a point closer to the main door of the chosen houses. In total, five of our ASTBFs captured 28 mosquitoes with a mean of 6.0 ± 1.5 (64.3% females and 37.7% males), while the BG-SBs captured a similar number of 30 mosquitoes with a mean of 10 ± 2.6 (63.3% females and 33.7% males). There was no significant difference in total catches between the UANL *Aedes* trap (ASTBF) and the BG-Sentinel trap was observed (*T-Welch* = −0.46, *df* = 1, *p* = 0.65). Likewise, there was no significant difference in the total capture of females (*T-Welch* = −0.24, *df* = 1, *p* = 0.81) and males (*T-Welch* = −0.27, *df* = 1, *p* = 0.78) between the UANL *Aedes* traps (ASTBF) and the BG-Sentinel traps (Table 2). With the data obtained from the field tests, we observed no difference in the number of captures of *Ae. aegypti* mosquitoes (regardless of sex) for both traps. Therefore, the UANL *Aedes* trap has an attraction percentage equal to the BG-Sentinel gold-standard attractant trap.

## 4. Discussion 

It has been documented that the chemical attractants evaluated in our study have demonstrated significant percentages of attraction with regard to *Ae*. *aegypti*, both individually and in blends [18]. The attraction results obtained in the laboratory and insectary varied depending on the attractant concentration and fan use. Blend 2 was the treatment selected to be used as the attractant bait for our trap since it presented the highest attraction rates in the HITSS (47.5 ± 4.8%) and had the highest spatial activity index (−0.45 ± 0.05), and these results were able to be replicated with consistency under larger flying spaces in entomological cages (40.8 ± 4.6%), including when using a fan to help disperse the attractant plumes. 

Volatility is the tendency of molecules to evaporate; the chemical nature of the attractant also explains this phenomenon [22]. Several authors have added battery or powered fans to enhance odor dispersal to exploit this property. In our 90 × 90 × 90 cm insectary cages, we noted that when a fan blows the wind, better attraction percentages were confirmed in our even larger 2 × 2 × 2 semi-field assays. Similarly, without a fan, where only the UANL *Aedes* trap^®^ (Blend 2: linalool, 15%; lactic acid, 10%; hexanoic acid, 1%; and ammonium chloride, 1%) was used, a capture percentage of 21.6 ± 3.5% was recorded 24 h after the first release; however, an increase of 50% was recorded (43.2 ± 4.0%) with the fan. How volatile attractants are released depends on the compound’s diffusion method, which is affected by the diffusion coefficient of the active ingredient and the product’s chemical properties. To ensure proper dispersion and protection of the product, techniques like microencapsulation, a suitable matrix, and fans are employed. These techniques consider the characteristics of the active ingredient, the releaser, and environmental factors [22]. 

In the semi-field assays, we found that the BG-Sentinel trap showed the best capture percentages for the three reading times (above 60%) but not for the laboratory tests, where only a 2 × 4 cm Whatman paper with its bait exposed was 8.3%, and with the fan, it was 34.8%. This confirmed for us the importance of proper bait formulation, proper trap design, and fan integration. On the other hand, for semi-field assays with only the pyramid cardboard body and the attractant sticky trap (AST), capture percentages between 10% and 40% were shown for the ASTB and ASTBF. We hypothesize that these percentages could be increased through a slow-release matrix of the attractant bait. A second factor to consider besides the fan is how to improve trap performance. However, a few complaints from dwellers about the trap’s electricity consumption were recorded. Future trap designs should avoid using fans to lower electricity consumption prices and to receive better acceptance by homeowners. On the other hand, although our results from the semi-field tests showed significant differences between the BG-SB and UANL *Aedes* trap (ASTBF) and favored the golden-standard BG-Sentinel, in the field tests, there was no statistically evident difference in the capture rates of *Ae. aegypti*; i.e., our UANL *Aedes* trap (ASTBF trap) captured the same number of mosquitoes as the gold-standard BG-SB in seven nights. These optimistic results motivate us to consider the possibility of a more extensive cluster randomized field trial that establishes new variables such as different seasons of the year and locations.

Vector population traps, such as GAT and BG-Sentinel, can be used to surveil *Aedes* species before and after outbreaks. Johnson et al. compared the use of these traps for gravid *Aedes* females and concluded that using them both is better [23]. GAT is more versatile and cost-effective than BG-Sentinel, which is more expensive and only sometimes well accepted. Our study shows the price difference between BG-Sentinel and our low-cost sticky trap made with over-the-counter products. Although this study has promising results, it is essential to consider a few things when conducting field tests. During a case-control study, Parra et al. faced many obstacles, including losing 660 traps and changing the location of 40 during a 36-week study. The leading causes of loss were inaccessible homes, holidays, and rainy days. Despite these setbacks, their traps captured 6024 *Aedes* and 1333 *Culex* mosquitoes. Similarly, our semi-field tests were affected by harsh weather conditions, but we could still conclude the study successfully [24].

In implementing the field assays, our UANL *Aedes* traps were well accepted by families. In contrast, the BG-Sentinel trap had to be relocated due to the discomfort caused by its smell. Our trap showed encouraging results for intra-home monitoring of *Ae. aegypti*. However, the design needs improvement for outdoor functionality. Our proposed low-cost entomological monitoring system can help surveil *Aedes* mosquito populations indoors and outdoors. This is crucial because standardized protocols for dengue surveillance rely on Aedic indices that are considered weekly estimators of transmission risk. Sometimes surveillance can be complemented with the use of ovitraps, which are sensitive enough to detect the presence of the vector, especially in areas where the level of infestation of *Ae. aegypti* is low, or they can be used to evaluate pupal indices as they are considered more accurate because most emerge as adults [25]. However, despite their great usefulness, larval indices do not always show a good correlation with the abundance of adult mosquitoes because a female can distribute her eggs across more than one breeding site. Pupal indices are rarely used due to the practical difficulties and labor of counting pupae, particularly those found in large containers [26]. According to Parra et al., traditional methods of measuring the risk of dengue infection in an area should not rely solely on immature stages of the mosquito, as only adult females can spread the virus through their bite. Instead, they suggest evaluating the entomological index obtained from an adult trap and comparing it with the cost of monitoring using the Breteau index (BI). Their results showed that the entomological index was positively correlated with the incidence of dengue, particularly during intervals with less intense vector control measures. Additionally, the operating costs of the adult index were lower than those of the BI, requiring 71.5% fewer human resources [24,26].

As a result, the number of alternative methods to estimate the abundance of adults or improvements to the available techniques is beginning to grow [27]. Actual vector demographic data are needed to monitor the study area and subareas, such as city blocks. The adult rates of *Ae. aegypti* described in the works of Parra et al. and Ong et al., as well as the use of attractive sticky traps such as the one developed in this study, can be applied at various levels of spatial aggregation for complete monitoring of the study area. However, although these indices and the use of adult traps are suitable for predicting dengue risk, they need to be tested and validated in various settings before routine use [24,27]. 

## 5. Conclusions

The development of our UANL attractant sticky trap (UANL *Aedes* Trap^®^) arises from the need to develop low-cost traps for the surveillance of adult mosquito *Aedes aegypti* populations in Latin America and other areas. Surveillance results can trigger a whole chain of medical care procedures and vector control activities. Conversely, a lack of or weak surveillance has severe consequences for health systems. *Ae. aegypti*, a daytime bitter species, is highly attracted to human skin compounds and can be captured using these compounds. However, the need to disperse the attractants using powered or battery fans increases costs. In this study, we used sticky cardboard surfaces and four attractive compounds (Blend 2: linalool, 15%; lactic acid, 10%; hexanoic acid, 1%; and ammonium chloride, 1%), along with a powered fan within a pyramid-shaped structure. Semi-field and field studies were conducted using the BG-Sentinel trap as a positive control, which is accepted as the golden standard. Interestingly, after a 7-day sampling period in the household environment, five of our prototypes captured 28 mosquitoes with a mean of 6.0 ± 1.5 (64.3% females and 37.7% males), while the BG-SB captured a similar number of 30 mosquitoes with a mean of 10 ± 2.6 (63.3% females and 33.7% males). No significant difference in total catches between the UANL *Aedes* trap (ASTBF) and the BG-Sentinel trap was observed. The results obtained in this work, under field and semi-field conditions, allow us to demonstrate the sensitivity and specificity of the trap, indicating its potential to be used in the future as a vector surveillance indicator, thereby replacing traditional and weak larval house indices to make dengue management recommendations.

## Figures and Tables

**Figure 1 insects-14-00940-f001:**
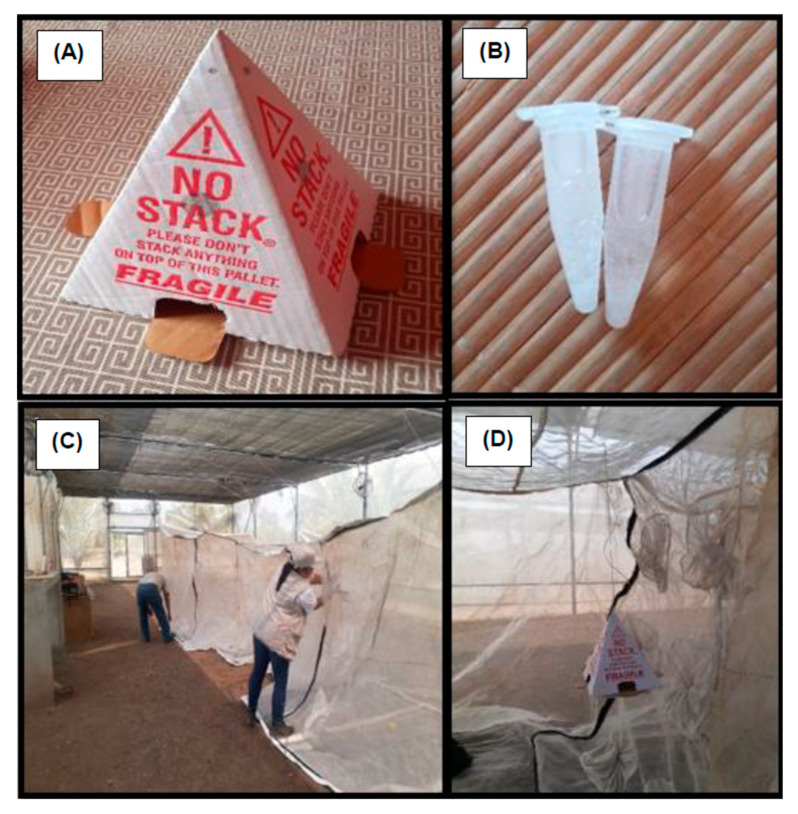
Setup for the semi-field experiments. (**A**) The UANL *Aedes* trap prototype is shown. (**B**) The attractant cartridge is positioned in the Eppendorf tube located on the lower left, and the fan powered by a 110 V source is positioned on the right. (**C**) The experiments were conducted in 2 × 2 × 2 m nylon-screened cages placed in a 10 × 50 × 5 m greenhouse. This greenhouse is located in the CRISP-INSP experimental field station situated in Ejido Río Florido, Tapachula, Chiapas. (**D**) The traps were elevated to a height of 60 cm and placed inside a five-row semi-field cage for the experiments.

**Figure 2 insects-14-00940-f002:**
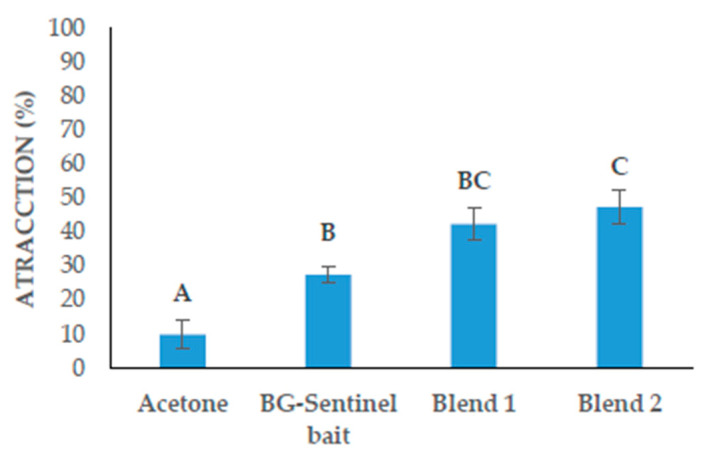
The results of screening using the HITSS to determine the attraction percentage of female *Ae. aegypti* mosquitoes in response to four different treatments. If two treatments share the same letter, it means there is no significant statistical difference between their means (*p* > 0.05).

**Figure 3 insects-14-00940-f003:**
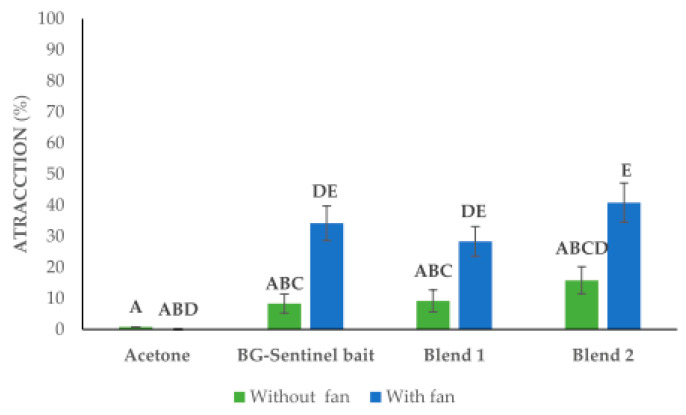
Attraction (%) of female *Ae. aegypti* mosquitoes in larger cages with a flight space measuring 90 × 90 × 90 cm. Treatments that share the same letters indicate that there is no significant statistical difference between their means (*p* > 0.05).

**Table 1 insects-14-00940-t001:** The percentage of captured female *Ae. aegypti* mosquitoes per trap 24 h and 7 days after the first release (100 mosquitoes) and 24 h after the second release (100 mosquitoes in 7 days).

Total Mosquitoes Captured Per Trap of 100 Female *Ae. aegypti* Released in Semi-Field Bioassays (*Percentage of Capture ± SE)*			
Trap System	24 h after the First Release	7 Days after the First Release	24 h after Second Release
Attractant sticky trap (AST)	9.6 ± 2.7%	12.8 ± 4.2%	2.4 ± 0.7%
Attractant sticky trap + Bait (ASTB)	21.6 ± 3.5%	27.0 ± 5.8%	10.0 ± 1.7%
Attractant sticky trap + Bait + fan (ASTBF)	43.2 ± 4.0%	*	40.8 ± 2.9%
BG-Sentinel + BG bait (BG-SB)	63.6 ± 11.1%	64.3 ± 11. 6%	65.3 ± 7.4%

***** Due to adverse weather conditions, which compromised the physical state of the traps if they were manipulated, the capture percentages could not be counted.

**Table 2 insects-14-00940-t002:** The total number of *Ae. aegypti* mosquitoes captured indoors by the UANL *Aedes* trap or attractant sticky trap (ASTBF) and BG-Sentinel trap (BG-SB) in different locations within Tapachula, Chiapas, Mexico.

**UANL Aedes trap (ASTBF)**			
	**Captured females**	**Captured males**	Total
Total	18	10	28
Average	3.6 ± 1.5	2.0 ± 1.0	6.0 ± 1.5
**BG-Sentinel trap** **(BG-SB)**			
Total	19	11	30
Average	6.3 ±4.5	3.7 ± 2.1	10.0 ± 2.6

Note: UANL *Aedes* trap baited with linalool, 15%; lactic acid, 10%; hexanoic acid, 1%; and ammonium chloride, 1%. BG-Sentinel trap baited with BG-Lure.

## Data Availability

All the data are contained within the article.

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
