# Peer review of "Design and Evaluation of a Sticky Attractant Trap for Intra-Domiciliary Surveillance of *Aedes aegypti* Populations in Mexico"

_insects, 2023, doi:10.3390/insects14120940_

Round 1

Reviewer 1 Report

Comments and Suggestions for Authors

Peer review report on the manuscript "Design and evaluation of an attractant sticky trap for surveillance of household indoors of female populations of Aedes aegypti in Mexico", (Manuscript ID: insects--2710034)

 Recommendation: Accept after minor revision

Comments to Authors:

This manuscript describes the design of a prototype low-cost sticky trap for indoor monitoring of adult populations of Aedes aegypti. It reports the results of its evaluation under laboratory and semi-field conditions. The effectiveness of the trap was tested either without a chemical attractant or with two different attractant compounds and compared with the effectiveness of the standard BG-Sentinel trap. The combination of the sticky trap with the attractant bait that resulted in higher performance was also evaluated in the field, and precisely, they were placed indoors into selected houses.

The laboratory and field trials were adequately designed and conducted, the results were sufficiently presented and statistically analyzed, and the discussion contributes to a better understanding of the results.

However, the manuscript suffers specific reporting weaknesses, and a revision is needed before it can be accepted as eligible for publication by Insects journal, considering the following comments.

 General Comments:

In the Materials and Methods, it needs to be clarified why and based on what data the two particular compounds (Blend 1 and Blend 2) were chosen as attractant baits to be combined and tested with the prototype trap. In paragraph 2.1. (lines 111-117) it is stated that the selection was made "Following higher documented results along with our lab experiences, …" but no experimental data or relevant literature reference is given.

In the description of the Attraction bioassay screening in the laboratory (Materials and Methods section, paragraph 2.2.), more information on the screening method and the counts taken should be given as the bibliographic reference (reference 19) on which the method is based is written in a different language (Spanish) from that of the Insects journal.

The experimental design of the field evaluation of UANL Aedes Trap in indoor household environments (paragraph 3.4.) is extremely weak as the two types of traps were tested on different environments under different conditions as the prototype trap with the attractant was placed indoors, and the reference trap outdoors. Even though this arrangement is justified by the authors with "The BG-Sentinel had to be relocated due to discomfort caused by its smell" (line 373), however, comparing the results from the captures of the two traps is questionable, and this should be clearly noted in the analysis and discussion.

The Conclusions section (lines 400-406) should better refer to conclusions drawn from the experiments of this study than to this generic comment.

and some Specific minor comments:

In lines 123 and 205: put a period at the end of the sentence.

In line 229: the word “mosquforoes” should be corrected

In line 259: change “.0001” to “0.001”

Reviewer 2 Report

Comments and Suggestions for Authors

This manuscript describes the development and evaluation of a novel attractant sticky trap for the collection of Aedes aegypti. The goal, to develop a simple and cheap surveillance trap is a worthy one and deserves exploration. Many current trap designs are too expensive for widespread use so a cheap alternative would be a huge benefit.

This is an interesting manuscript and presents some good information but has numerous issues which must be addressed before publication. Some issues are methodological, and some are analytical but addressing both types are critical to properly evaluate the results and conclusions. The manuscript also needs a good bit of editing polish to improve readibility. Much of the included information is very good, but it needs to be presented more directly.

Specific comments are below.

L1-4: The title is comprehensive but is difficult as written. Suggest something soimpler like: D & E of of an attractant sticky trap for surveillance of indoor populations of Aedes aegypti in Mexico.

L22-24 & L91-93; Sentence is difficult. Information is fine, but this does not read well.

L28: Suggest, "evaluate a low-cost, attractive sticky trap..."

L29-31: Sentence is difficult.

L61: Suggest "Zika virus cases" after deletion of "the disease caused by."

L64-66: Suggest remval of "On the other hand, the evidence that" be replaced with "Evidence supported Aedes aegypti as the primary vector...."

L75-76: What is "alarm data?" Please explain in text.

L77-80: Sentences here can be combined into one clear sentence.

L116-117: Is there a patent or patent application for this trap? If so, it should be declared.

L125: What range of concentrations were applied? The volume was 50ul but what concentration or quantity as stated for the BG-Lure in L134?

Section 2.2: Clear statement of controls (positive and negative) as well as number of replicates. What data was collected at the end of the 20min exposure? The text doesn't say.

L140: "strips"

L144: How and why were mosquitoes removed?

L164: Figure 1 legend needs revision. The figure shows the UANL trap prototype and the semi-field experimental setup.

Figure 2: L244-249: The results show that Blend1 and Blend 2 were not significantly different from one another. So why was blend2 chosen for further testing?

Why are the treatments called by statistical group rather than just using the treatment name? Clearer to say, "the results were significantly different than acetone and BG Sentinel bait treatments."

Figure 3, L265-273: Same comment about terminology as for Figure 2. I am concerned about the result of the analysis with fan. Shouldn't BG-S bait w/ fan also be group DEF? It has a mean and SE that put it in the middle between Blend1 w/ fan and Blend 2 w/ fan. If 28.3 and 40.8 are both in group F then wouldn't a value between the two also be in group F? Something seems incorrect. Please check these stats.

Table 1: Suggest this be presented as either a summary table or as a figure.

L300-310: This appears to say that ASTBF were placed in the living room or bedroom, but BG-SB were placed in yard (outside?) or hallway. If so, not sure that this is the best comparison for analysis. Clearly yard/hallway could be very different than LR/BR so the counts could be measuring very different things. This could well confound the results. This needs to be thoroughly addressed in the methods.

Table 2: Same as above. Better as a summary table or figure.

Comments on the Quality of English Language

The manuscript needs thorough editing. Some specific comments are noted above but these are representative, not exhaustive.

Round 2

Reviewer 2 Report

Comments and Suggestions for Authors

This revised version has addressed the issues raised at first review. It is much improved. Only a couple of very minor comments on the current version are below.

L43: "Interesting" is a very subjective word. Suggest just dropping it and starting with "Results...."

L439-444: The data availability statement for this manuscript is missing.

Author Response

L43: "Interesting" is a very subjective word. Suggest just dropping it and starting with "Results...." DONE, highlighted in yellow

L439-444: The data availability statement for this manuscript is missing.

DONE: highlighted in yellow